# Adverse events associated with cyclophosphamide: A pharmacovigilance study using the FDA adverse event reporting system

Hui Zhang[1], Minghao Lin[1], Dexi Zhao[1,2], Yujuan Fu [1*]

1 Changchun University of Chinese Medicine, Changchun, China, 2 The Affiliated Hospital to Changchun University of Chinese Medicine, Changchun, China

* fuyujuan1111@163.com

## Abstract

### Objective

Cyclophosphamide(CHOP) is a widely used chemotherapeutic agent belonging to the class of alkylating agents. It primarily inhibits the proliferation of tumor cells by interfering with DNA replication and transcription. CHOP has been approved for the treatment of various types of malignant tumors. The aim of this research is to assess adverse events linked to CHOP using real-world data.

### Methods

This research collected and analyzed adverse event reports associated with CHOP from the FAERS(US Food and Drug Administration Adverse Event Reporting System) database spanning from the first quarter of 2004 to the third quarter of 2024. This study leveraged the FAERS database, extracting reports submitted exclusively by healthcare professionals where CHOP was designated as the primary suspect (PS). Four risk signal detection methods were employed: Reporting Odds Ratio (ROR), Proportional Reporting Ratio (PRR), Bayesian Confidence Propagation Neural Network (BCPNN), and Multiitem Gamma Poisson Shrinker (MGPS), to conduct a comprehensive analysis of CHOP-related ADRs.

### Results

A total of 3,625 adverse event reports with CHOP as the primary suspect drug were obtained. Across 25 system organ classes (SOCs), four positive Preferred Terms(PTs) signals were detected. The most significant signal in SOCs was "blood and lymphatic system disorders" (ROR = 6.64, 95% CI 6.49–6.79), while the most significant PTs signal was "high grade B-cell lymphoma Burkitt-like lymphoma recurrent" (ROR = 613.63, 95% CI 123.85–3040.40).

**Data availability statement:** All relevant data are within the paper. We confirm that my submission contains all raw data required to replicate the results of my study.

**Funding:** This work was supported by the Jilin Provincial Science and Technology Department Project (YDZJ202401092ZYTS). Funding Program: Jilin Provincial Department of Education: JJKH20230958KJ.

**Competing interests:** The authors have declared that no competing interests exist.

## Conclusion

This study provides compelling evidence of the presence of CHOP seven unexpected off-label AEs were also observed, such as mucoepidermoid carcinoma, hypotelorism of orbit, breakthrough COVID-19, airway remodeling, meningitis enteroviral, acute graft versus host disease, and gastrosplenic fistula. Additionally, the combination of CHOP and radiotherapy increased the risk of several severe AEs, such as hemorrhagic cystitis. These findings highlight the importance of caution when prescribing CHOP to high-risk individuals with a history of meningitis, cystitis or airway obstruction.

## Introduction

Cyclophosphamide (CHOP) is a widely used chemotherapeutic agent belonging to the nitrogen mustard alkylating agents. Since its introduction in the 1950s, CHOP has become one of the cornerstone drugs in the treatment of various malignant tumors and autoimmune diseases [1]. After being metabolized in the liver, it is converted into cytotoxic active substances, which inhibit the proliferation of tumor cells by cross-linking with DNA, thereby achieving an anti-tumor effect [2]. The clinical application of CHOP is extensive, covering hematological malignancies (such as non-Hodgkin's lymphoma, multiple myeloma, leukemia, etc.) and solid tumors (such as breast cancer, ovarian cancer, small cell lung cancer, etc.). It is metabolized into active forms in the body through liver cytochrome P450 enzymes (such as CYP2B6, CYP3A4, etc.). The main metabolic processes include: 4-hydroxyCHOP: the main active metabolite of CHOP; aldophosphamide: 4-hydroxyCHOP further decomposes into aldophosphamide; phosphoramide mustard and acrolein: aldophosphamide decomposes into cytotoxic phosphoramide mustard and acrolein within cells [3]. DNA cross-linking disrupt the double helix structure of DNA, hindering DNA replication and transcription, thereby inhibiting the proliferation of tumor cells [4]. Cross-linking also triggers DNA damage repair mechanisms, and when repair fails, cells initiate the apoptsosis program. In addition to its immunosuppressive effects, CHOP is also used to treat autoimmune diseases such as systemic lupus erythematosus and rheumatoid arthritis [5]. Although CHOP has demonstrated significant efficacy in clinical practice, its use is also associated with certain side effects, such as myelosuppression, alopecia, nausea, and vomiting. Long-term use may also increase the risk of secondary tumors. Therefore, in recent years, research has gradually shifted towards improving the targeting of CHOP, reducing side effects, and overcoming tumor resistance [6]. The application of novel drug delivery systems, combination therapy strategies, and gene editing technologies has provided new directions for the optimized use of CHOP [7].

   While CHOP has exhibited notable effectiveness and activity, it necessitates vigilant attention to safety and the surveillance of specific toxic reactions. The FAERS, which is prevalently employed for adverse event (AEs) signal detection studies, serves as a valuable tool for monitoring and assessing the post-marketing safety of medications. Compared with previous studies, we used FAERS data mining

technology to retrospectively detect and analyze the adverse reaction signals of CHOP, and further expanded the time span of the study.

## 2 Materials and methods

### 2.1 Data source

All our data comes from the FAERS Database (https://fis.fda.gov/extensions/FPD-QDE-FAERS/FPD-QDE-FAERS.html) from the first quarter of 2004 to the third quarter of 2024. All data were from the public database, and no ethical approval was needed. This system consists of adverse event and medication error reports voluntarily submitted to the FDA by healthcare professionals and consumers. The database comprises seven subsets: patient information (DEMO), drugs (DRUG), indications (INDI), outcomes (OUTC), adverse reactions (REAC), medication timing (THER), and reporting country (RPSR). These subsets are linked and analyzed through the personal safety report code (primaryid) field. The data processing and analysis software used in this study is R software. This observational analysis utilizes the FDA Adverse Event Reporting System (FAERS) database, which is updated quarterly and includes self-reported data from healthcare professionals (doctors, pharmacists, healthcare specialists, and registered nurses) and non-healthcare professionals (consumers, lawyers, sales representatives, etc.). The FAERS database has been widely used to identify potential adverse drug reactions. It includes unique identifiers, reporting dates, reporting country/region, primary reporter qualifications, patient demographic information (such as gender, age, and weight), suspected and concomitant drugs and their indications, ADR occurrence date, and ADR manifestations.

### 2.2 Data cleaning

Since the FAERS database can be reported by different institutions and the public, there are multiple versions of duplicate cases and non-standardized data. Therefore, before analysis, it is necessary to remove duplicates using the primaryid and caseid in the seven files. The primaryid is the unique number for identifying FAERS reports and the main linking field in the data files, while the caseid is the number for identifying cases. Each caseid corresponds to one Individual Case Safety Report (ICSR). When an ICSR is submitted multiple times subsequently, its caseid remains unchanged, but the primaryid will generate a new, larger number.

### 2.3 Data extraction and standardization

Data from the FAERS Database from 2004Q1 to 2024Q3 were extracted using 'CHOP' as the search term, and duplicate reports from the same patient were removed. The collected data mainly included the following characteristics of the drug-related reports: patient age, patient gender, type of reporter, adverse event outcomes (death, life-threatening, hospitalization, disability, and/or other), medications, and indications. The data were standardized using the preferred terms (PTs) from version 26.1 of the Medical Dictionary for Regulatory Activities (MedDRA), and adverse events were categorized into different systems using the system organ class (SOCs).

### 2.4 Data statistical analysis

Summary of FDA-approved CHOP, as is shown in **Table 1**. To reduce the probability of false positive signals, this study used four methods to evaluate the correlation between drugs and adverse reaction events: Reporting Odds Ratio (ROR),

**Table 1. Summary of FDA-approved Cyclophosphamide.**

| Generic name | Brand name | Approval date |
|---|---|---|
| Cyclophosphamide | CYTOXAN, ENDOXAN, LYOPHILIZED CYTOXAN | 1959 |

Proportional Reporting Ratio (PRR), Bayesian Confidence Propagation Neural Network (BCPNN), and Multiitem Gamma Poisson Shrinker (MGPS). The higher the signal value of the adverse event, the stronger the signal, indicating a stronger correlation between the drug and the target adverse event. The calculation formulas and evaluation criteria for the four methods are listed in **Table 2**. All data cleaning, statistical analysis, and data visualization were performed using R software (version 4.4.0).

## 3 Results

### 3.1 Descriptive analysis

A total of 28,591 reports were obtained for CHOP, with 13,364 from males, 7,910 from females, and 7,317 unknown, as shown in **Fig 1**. The majority of patients were aged 41–65 years and above, accounting for 2,640 cases. The main reporting countries were the United States, France, China, Canada, and Italy, as shown in **Fig 2**. The majority of reports were submitted by Physicians, accounting for 38.5%. The most common outcome was other serious (45.34%), followed by hospitalization (28.55%). The year with the highest number of reports was 2014, as shown in **Fig 3**. Adverse reactions mainly occurred within <7 days and 7–28 days, as shown in **Fig 4**.

### 3.2 Disproportionality analysis

After screening CHOP, we identified a total of 83,576 positive risk signals for adverse events (AEs), spanning across 25 SOCs, as detailed in **Table 3**. A comparison with the existing CHOP product label reveals that the primary affected organ systems are categorized based on SOCs classification. Notably, blood and lymphatic system disorders emerged as the

**Table 2. Four major methods used for signal detection.**

| Method | Formula | Threshold |
|--------|---------|-----------|
| ROR | $ROR = ad/bc$ | $a \geq 3$ |
|  | $95\%CI = e^{\ln(ROR)\pm 1.96(1/a+1/b+1/c+1/d)^{0.5}}$ | $ROR \geq 2$ |
|  |  | 95%CI (lower limit) > 1 |
| PRR | $PRR = a(c+d)/c/(a+b)$ | $a \geq 3$ |
|  | $\chi^2 = [(ad-bc)^2](a+b+c+d)/[(a+b)(c+d)(a+c)(b+d)]$ | $PRR \geq 2$ |
|  |  | 95%CI (lower limit) > 1 |
| BCPNN | $IC = \log_2 a(a+b+c+d)/(a+c)/(a+b)$ | IC025 > 0 |
|  | $95\%CI = E(IC) \pm 2V(IC)^{0.5}$ |  |
|  | $r = (a+b+c+d)^2/(a+b+1)/(a+c+1)$ |  |
|  | $E(IC) = \log_2 a(a+b+c+d)^2/(a+b+c+d+r)/(a+b)/(a+c)$ |  |
|  | $V(IC) = 1/\ln2(b+c+d+r-1)/(a+1)/(a+b+c+d+r+1)+(2+b+c+2d)/(a+b+1)/(a+b+c+d+r+3)$ |  |
|  | $IC025 = E(IC) - 2V(IC)^{0.5}$ |  |
| EBGM | $EBGM = a(a+b+c+d)/(a+c)/(a+b)$ | EBGM05 > 2 |
|  | $95\%CI = e^{\ln(EBGM)\pm 1.96(1/a+1/b+1/c+1/d)^{0.5}}$ |  |
|  | $EBGM05 = e^{\ln(EBGM)-1.96(1/a+1/b+1/c+1/d)^{0.5}}$ |  |

Formula: a, number of reports containing both the suspect drug and the suspect adverse drug reaction; b, number of reports containing the suspect adverse drug reaction with other medications (except the drug of interest); c, number of reports containing the suspect drug with other adverse drug reactions (except the event of interest); d, number of reports containing other medications and other adverse drug reactions. ROR, reporting odds ratio; CI, confidence interval; N, the number of co-occurrences; PRR, proportional reporting ratio; $\chi^2$, chi-squared; BCPNN, Bayesian confidence propagation neural network; IC, information component; IC025, the lower limit of the 95% one-sided CI of the IC; EBGM, empirical Bayesian geometric mean; EBGM05, the lower 95% one-sided CI of EBGM.

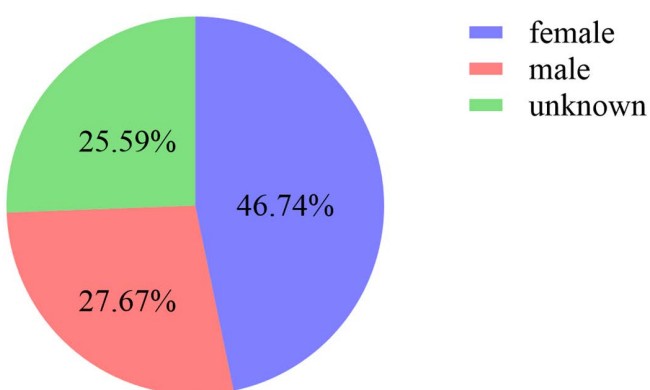

**Fig 1. Proportion of adverse drug outcomes.**

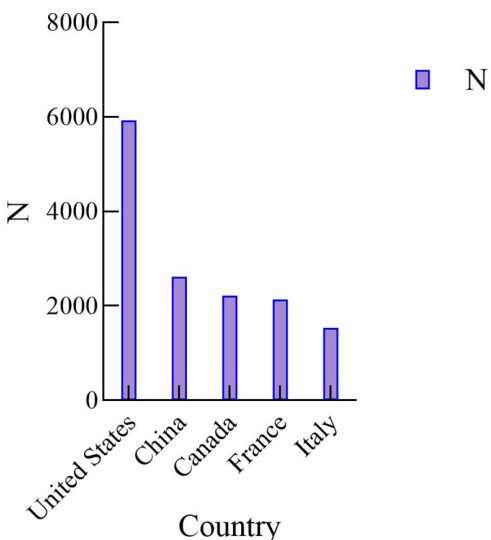

**Fig 2. Reporter country.**

most prominent category, with 729 positive risk signals specifically for blood-related AEs. Among these, high-grade B-cell lymphoma Burkitt-like lymphoma recurrent exhibited the strongest positive signal, with a relative risk (RR) of 613.53 [95% CI: 123.85, 3040.4]. According to the ranking by ROR, the main systems affected by CHOP are blood and lymphatic system disorders, neoplasms benign, malignant and unspecified (including cysts and polyps), infections and infestations, and hepatobiliary disorders. The high-intensity PTs are high grade B-cell lymphoma Burkitt-like lymphoma recurrent RR [613.53, 95% CI (123.85, 3040.40)], precursor T-lymphoblastic lymphoma/leukAEsmia recurrent RR [327.29, 95% CI (138.76, 771.98)], and T-cell lymphoma recurrent RR [232.22, 95% CI (125.54, 429.54)], as shown in **Fig 5**.

## 4 Discussion

### 4.1 CHOP 20 year span analysis

This study maintains a certain continuity with previous studies in terms of data sources and analysis methods, both being based on the FDA Adverse Event Reporting System (FAERS) database for drug safety signal mining [8]. However,

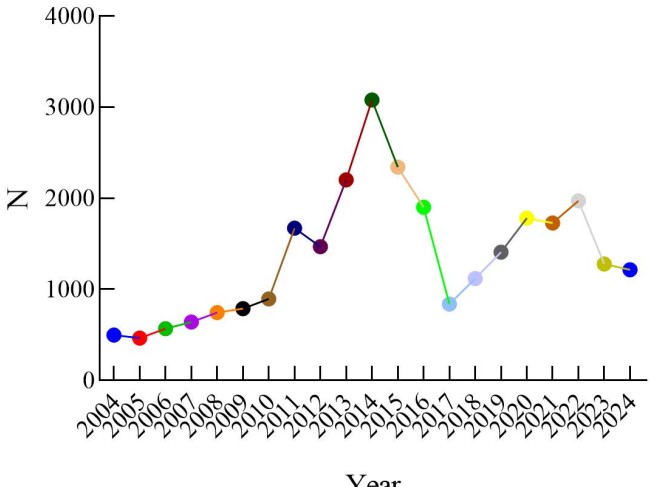

**Fig 3. Distribution of adverse events due to CHOP reported annually.**

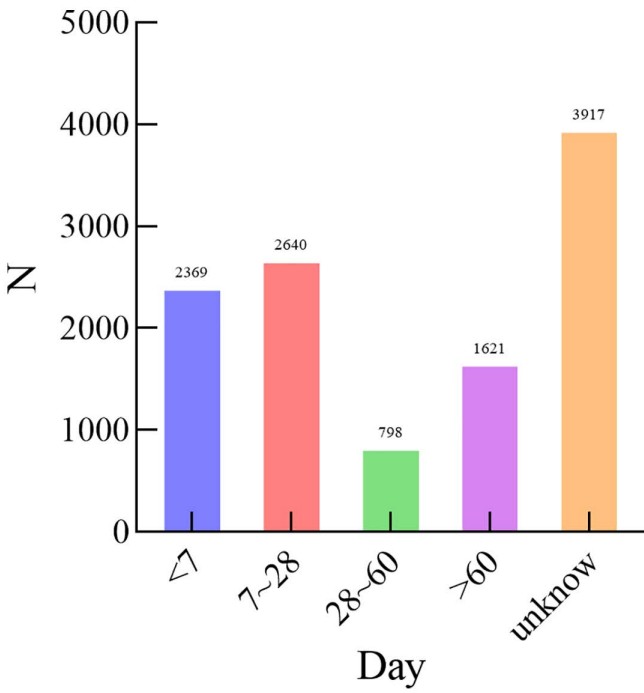

**Fig 4. Time of adverse events due to CHOP reported annually.**

compared with previous studies, this study has the following significant advantages: First, in terms of data time span, this study includes data from 2004Q1 to 2024Q3, a total of 20 years, which is a significant improvement over the 3–5 year data span commonly used in previous studies. This long-term data collection not only captures more rare adverse events but also allows for the observation of the dynamic changes in drug safety signals, providing a more comprehensive temporal dimension for drug safety evaluation. Second, in terms of data volume, the increase in sample size significantly

**Table 3. Disproportionate analysis in top 4 SOC.**

| soc_english | Case reports | ROR(95% CI) | PRR(95% CI) | chisq | IC(IC025) | EBG-M(EBGM05) |
|---|---|---|---|---|---|---|
| blood and lymphatic system disorders | 8771 | 6.64(6.49, 6.79) | 6.05(5.93, 6.17) | 37234.38 | 2.58(2.55) | 6(5.89) |
| neoplasms benign, malignant and unspecified (incl cysts and polyps) | 5487 | 2.49(2.42, 2.56) | 2.39(2.34, 2.44) | 4550.42 | 1.25(1.22) | 2.39(2.33) |
| infections and infestations | 9174 | 2.16(2.11, 2.21) | 2.03(1.99, 2.07) | 5070.07 | 1.02(0.99) | 2.03(1.99) |
| hepatobiliary disorders | 1588 | 2.04(1.94, 2.15) | 2.02(1.94, 2.1) | 825.18 | 1.01(0.94) | 2.02(1.94) |

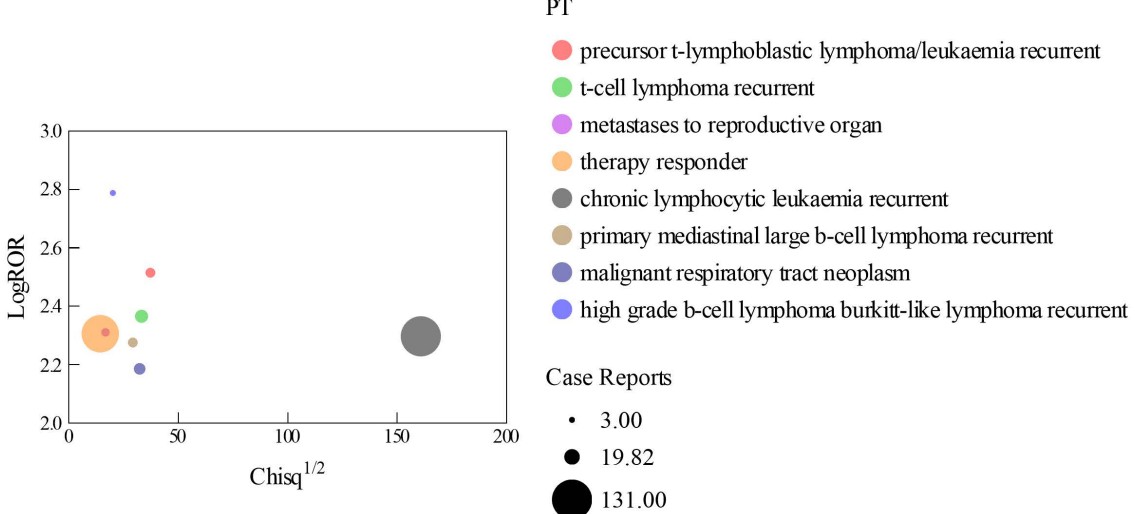

**Fig 5. Bubble diagram of the PTs.**

enhances the power of statistical analysis, aiding in the discovery of more potential safety signals. Moreover, the long-term data can better control the interference of seasonal factors on the analysis results, enhancing the reliability of the study conclusions. Lastly, this long-term study design enables us to observe the safety evolution of newly marketed drugs and the potential risk changes brought by the repurposing of old drugs, providing a more comprehensive reference for clinical medication safety [9].

## 4.2 Basic situation of CHOP-related adverse event reports

The study results show that the number of adverse event reports for CHOP is related to gender. Among the CHOP reports with known gender, there are more reports from females than males, suggesting that when CHOP is used in female patients, there should be a heightened vigilance for the occurrence of adverse events. In terms of report age, the age group most affected by adverse events is 41–65 years old. However, there are still 11,912 reports with unknown age in this study, which may have a certain impact on the results. The top 5 countries in terms of reporting numbers are the United States, France, China, Canada, and Italy, with the United States having the highest number of reports. This may be related to the country where the database developer is located, the time of market availability in various countries, the country of origin of the drug, and the awareness, dissemination, and emphasis on drug adverse reactions among people in different countries. However, there are still 1,318 reports with unknown reporting countries in this study, which may have a certain impact on the results.

### 4.3 CHOP-related AEs involvement in SOCs

CHOP reports adverse events involving 25 SOCs in the MedDRA classification, which may be related to the broad scope of its mechanism of action. Among them, blood and lymphatic system disorders, neoplasms benign, malignant and unspecified (including cysts and polyps), infections and infestations, and hepatobiliary disorders have a higher number of reports and stronger signals, consistent with the descriptions in the product insert. This confirms the credibility of the methods used in this study.

### 4.4 CHOP-related adverse events ranked by report number

Among the 2,671 PTs, the highest number of positive reactions was found for high grade B-cell lymphoma Burkitt-like lymphoma recurrent. Current research indicates that CHOP is largely consistent with the product insert for precursor T-lymphoblastic lymphoma/leukAEsmia recurrent, T-cell lymphoma recurrent, metastases to reproductive organ, therapy responder, and chronic lymphocytic leukAEsmia recurrent. This phenomenon may be related to the central therapeutic role of the drug in aggressive B-cell malignancies: as a component of classic chemotherapy regimens such as CHOP, cyclophosphamide exerts a significant cytotoxic effect on highly proliferative Burkitt-like lymphoma cells by alkylating and disrupting DNA structure. It is worth noting that the concentration of positive reactions observed in this study may also reflect the urgent clinical need for treatment regimens for recurrent/refractory Burkitt lymphoma.

### 4.5 CHOP and other suspected PTs signals

The study results show that the PTs for blood and lymphatic system disorders caused by CHOP include lymphoid tissue hypoplasia and acquired von Willebrand's disease. This suggests that when using CHOP clinically, there should be vigilance for adverse reactions in the blood system [10]. Considering that the indications for CHOP overlap with certain PTs symptoms in the tumor system, hepatobiliary system diseases, blood system, and infections, and that changes in related examination indicators may be due to the disease being treated with CHOP itself, or due to the effects of CHOP or subsequent infections, clinical judgment should be exercised during application [11]. Therefore, during the treatment of hepatobiliary and central nervous system diseases, clinicians should carefully identify whether adverse events are present [12]. Additionally, seven unexpected off-label AEs were observed, such as mucoepidermoid carcinoma, hypotelorism of orbit, breakthrough COVID-19, airway remodeling, meningitis enteroviral, acute graft versus host disease, and gastrosplenic fistula. Immune suppression related events (e.g., breakthrough COVID-19, enterovirus meningitis) may be related to the profound lymphocytopenia induced by cyclophosphamide; Abnormal tissue repair (e.g., airway remodeling, gastro-splenic fistula) may reflect the drug's interference with fibroblast function; Secondary malignancies (e.g., mucoepidermoid carcinoma) require long-term follow-up to determine whether they are related to the carcinogenic risk of alkylating agents. The FAERS data cannot establish causality, and the sample size of off-label AEs is limited. Future prospective studies are needed to verify their associations. It is recommended to improve pharmacogenomic testing (such as CYP2B6 metabolizing enzyme analysis) in clinical practice to optimize the individualized medication strategy for cyclophosphamide. CHOP therapy exerts its antitumor effect by killing rapidly proliferating cells, but its nonspecific inhibition of lymphocytes (especially T cells, B cells, and NK cells) not only leads to a reduction in their quantity but also induces qualitative damage to immune function [13]. The killing of memory lymphocytes by CHOP renders patients unable to establish long-term protective immunity following infection, increasing the risk of reinfection; meanwhile, in the immune-suppressed state, COVID-19 may progress to severe illness (e.g., acute respiratory distress syndrome [ARDS]). This mechanism is associated with virus-induced "cytokine storm" — lymphopenia fails to effectively regulate the inflammatory response, but instead promotes the massive release of pro-inflammatory cytokines such as IL-6 and TNF-α, exacerbating lung injury [14]. Currently, there are no large-scale epidemiological studies directly linking cyclophosphamide to the development of mucoepidermoid carcinoma (MEC). However, case reports suggest that immunosuppressed patients (such as those using CHOP after transplantation) may

have an increased risk of salivary gland tumors. MEC is commonly found in the salivary glands (such as the parotid gland), and the association between alkylating agent exposure and head and neck tumors needs further validation.

## 4.6 Mechanism of action of CHOP

The mechanism of action involves inhibiting cell proliferation by blocking DNA synthesis and nonspecifically killing lympho-cytes, thereby exerting immunosuppressive and anti-inflammatory effects [15]. CHOP has been used for over 50 years in the treatment of lupus nephritis (LN) and is the most commonly used first-line drug for severe LN [16]. CHOP undergoes primary bioactivation in the liver via the mixed-function microsomal oxidase system, resulting in the formation of active alkylating metabolites. These metabolites disrupt the growth of rapidly proliferating malignant cells that are susceptible to their effects [17]. The liver serves as the primary site for CHOP activation, with approximately 75% of the administered dose being activated by hepatic microsomal cytochrome P450 enzymes, including CYP2A6, 2B6, 3A4, 3A5, 2C9, 2C18, and 2C19. Among these, CYP2B6 demonstrates the highest 4-hydroxylase activity. The activation of CHOP leads to the formation of 4-hydroxyCHOP, which exists in equilibrium with its open-ring tautomer, aldophosphamide. Both 4-hydroxyCHOP and aldo-phosphamide can be oxidized by aldehyde dehydrogenase to produce inactive metabolites: 4-ketoCHOP and carboxyphos-phamide, respectively. Aldophosphamide can also undergo β-elimination to yield active metabolites such as phosphoramide mustard, acrolein, and nitrogen mustard. This spontaneous transformation can be facilitated by albumin and other proteins. Numerous in vitro and in vivo genotoxicity studies have demonstrated that CHOP possesses mutagenic and clastogenic properties [18]. Some studies have indicated that patients who developed cardiotoxicity received doses ranging from 120 to 270 mg/kg body weight within 1–8 days [19]. Low doses (e.g., < 2 g/m² per administration): mainly characterized by myelo-suppression, which is usually manageable. High doses or cumulative doses (e.g., > 6.5 g/m² or total dose >30 g): vigilance is required for irreversible organ damage and long-term secondary cancers. When used clinically, the balance between efficacy and toxicity must be strictly weighed, and the treatment regimen should be adjusted according to the patient's condition.

## 4.7 High-risk signal analysis

Zhang Xiwen [20] reviewed the reproductive toxicity of CHOP, finding that it has strong reproductive toxicity in both males and females. Yang Mingyue [21] reported a case of cardiotoxicity caused by conventional-dose CHOP in a child with leu-kemia. Literature [22] indicates that in addition to reproductive toxicity, CHOP can also lead to osteoporosis and intestinal mucosal damage, with conclusions mostly derived from animal experiments and lacking clinical data support. Currently, there are no published studies on the adverse reactions of CHOP based on big data [23–24]. The high-frequency PTs results statistically analyzed in this chapter are consistent with the drug instructions, indicating the reliability of the study results [25]. CHOP mainly causes adverse events such as febrile neutropenia, fever, death, nausea, and decreased white blood cell count, which are consistent with the reported results that CHOP often induces blood and lymphatic system dis-eases and gastrointestinal reactions [26].

This study still has certain limitations. First, the FAERS database is a spontaneous reporting system, and the data have issues such as underreporting, misreporting, and missing information, with diverse sources of information and potential reporting bias. Second, some patients' drug ADRs may be caused by complications or comorbidities. In clinical treatment, CHOP is often used in combination with other drugs, and the causal relationship between drugs and ADEs still needs fur-ther clinical research and evaluation. Additionally, the data in this study mainly come from European and American coun-tries. Due to differences in ethnicity and region, the results may have certain deviations in China.

## 5 Conclusions

This study, based on the FAERS database, utilized the ROR method and BCPNN method to mine signals of adverse events related to CHOP, providing new supplements to the relevant adverse reactions in its drug instructions with more real – world data. In future clinical applications, it is necessary to strengthen the management of the standardized clinical

use of CHOP and the management of off – label drug use. Medical staff should work together to judge the changes in relevant laboratory indicators during clinical application. Although the side effects of cyclophosphamide are known, its risk may be amplified in the pathological state of hematological malignancies. Therefore, the occurrence of adverse events is the result of the combined effect of 'drug effect - disease background - treatment strategy' rather than a single factor. CHOP should be used with caution in patients with hematological tumors, infections, and hepatobiliary diseases. At the same time, vigilance should be maintained against relevant adverse reactions. Patients should be informed in advance of the possibility of adverse reactions such as myelosuppression, immunosuppression, and infections. The necessity of routine blood cell counts should be explained. Patients should be guided to monitor their body temperature regularly and report any fever immediately. Vigilance should be maintained against situations such as hematuria and respiratory tract infections. Women of childbearing age should avoid pregnancy during the medication period as it can easily cause fetal malformations. Side effects such as nausea, vomiting, stomatitis, poor wound healing, amenorrhea, premature menopause, infertility, and alopecia may be related to the administration of CHOP. Prudent consideration should be given to the use of this drug. In case of adverse reactions, patients should report to the doctor in a timely manner for treatment.

## Supporting information

**S1 File. Descriptive analysis of Cyclophosphamide.**
(XLSX)

**S2 File. Disproportionate analysis in SOC.**
(XLSX)

## Author contributions

**Investigation:** Hui Zhang.

**Resources:** Hui Zhang, Dexi Zhao, Yujuan Fu.

**Software:** Hui Zhang, Minghao Lin.

**Writing – original draft:** Hui Zhang, Minghao Lin.

**Writing – review & editing:** Dexi Zhao, Yujuan Fu.

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
