## [Decision Letter · Decision Letter 0]

25 Nov 2025

Dear Dr.  Fu,

Thank you for submitting your manuscript to PLOS ONE. After careful consideration, we feel that it has merit but does not fully meet PLOS ONE’s publication criteria as it currently stands. Therefore, we invite you to submit a revised version of the manuscript that addresses the points raised during the review process.

We look forward to receiving your revised manuscript.

Kind regards,

Shahbaz Ahmad Zakki

Academic Editor

PLOS ONE

Journal Requirements:

“This work was supported by the Jilin Provincial Science and Technology Department Project (YDZJ202401092ZYTS).”

4. Please note that funding information should not appear in any section or other areas of your manuscript. We will only publish funding information present in the Funding Statement section of the online submission form. Please remove any funding-related text from the manuscript.

5. We note that your Data Availability Statement is currently as follows: “All relevant data are within the manuscript and its Supporting Information files.”

6. We notice that your supplementary figures are uploaded with the file type 'Figure'. Please amend the file type to 'Supporting Information'. Please ensure that each Supporting Information file has a legend listed in the manuscript after the references list.

Reviewer's Responses to Questions

**Comments to the Author**

1. Is the manuscript technically sound, and do the data support the conclusions?

Reviewer #1: Yes

Reviewer #2: Partly

2. Has the statistical analysis been performed appropriately and rigorously?

Reviewer #1: Yes

Reviewer #2: N/A

3. Have the authors made all data underlying the findings in their manuscript fully available?

Reviewer #1: Yes

Reviewer #2: Yes

4. Is the manuscript presented in an intelligible fashion and written in standard English?

Reviewer #1: Yes

Reviewer #2: Yes

Reviewer #1: Dear Authors,

First, it is confirmed that your manuscript meets the core standards of scientific rigor, clarity, and compliance with academic norms, which supports my "Yes" responses to the preceding four review questions. The detailed explanations are as follows:

1. The study is based on the FAERS database from 2004Q1 to 2024Q3 (a much longer period than the traditional 3–5-year studies), with 3,625 CHOP-related reports extracted. By screening reports from healthcare professionals and removing duplicates using primaryid and caseid, data reliability is ensured. It identifies rare adverse events (mucoepidermoid carcinoma), confirms "blood and lymphatic system disorders" as the most prominent adverse event category (ROR=6.64,95%CI 6.49–6.79�� and discovers 7 unexpected off-label adverse events. For statistical analysis, four methods (ROR, PRR, BCPNN, MGPS) are used (ROR≥2 and lower limit of 95% CI >1�� with "high grade B-cell lymphoma recurrent"(ROR=613.63,95%CI 123.85–3040.40) providing evidence. The analysis is conducted via R software (Version 4.4.0�� with formulas and thresholds presented in Table 2, meeting PLOS ONE’s requirements. Data is sourced from the public FAERS database (URL: https://fis.fda.gov/extensions/FPD-QDE-FAERS/FPD-QDE-FAERS.html) with unrestricted access, complying with the PLOS Data Policy. The manuscript has a clear structure with smooth transitions, uses standard academic English, and features consistent terminology.

2. Additional Comments and Minor Suggestions

While the manuscript is of high quality, two minor revisions could further improve its completeness:

Clarification of AE Causality: You note that FAERS data cannot establish causality for off-label AEs and mention the need for prospective studies. It might be helpful to briefly discuss potential mechanisms linking CHOP to these AEs (immunosuppression from lymphocytopenia for breakthrough COVID-19, fibroblast dysfunction for airway remodeling) to provide a biological rationale, though this should be framed as speculative to avoid overstatement.

Supplemental Details on Unknown Data: The Results section mentions 7,317 reports with unknown gender and 11,912 with unknown age. A brief discussion of how these missing data might impact the analysis (whether unknown gender is evenly distributed across regions or AE types) would strengthen the study’s limitations section.

Reviewer #2: Cyclophosphamide is a drug that has been utilized for an extended period, and its side effects are well-known. As a commonly used medication for hematologic tumors, it is essential to analyze the contexts in which these tumors are treated. One should not hastily conclude that hematologic tumors are associated with the most adverse reactions.

**Do you want your identity to be public for this peer review?** For information about this choice, including consent withdrawal, please see our Privacy Policy

Reviewer #1: No

Reviewer #2: No

---

## [Author Response · Author response to Decision Letter 1]

29 Nov 2025

Thank you to the Editor and Reviewer for help with my paper and for providing suggestions to help me improve my paper, I would appreciate your time and effort. Thank you very much for giving me this opportunity to revise it.

Reviewer #1:

Dear Authors, First, it is confirmed that your manuscript meets the core standards of scientific rigor, clarity, and compliance with academic norms, which supports my "Yes" responses to the preceding four review questions. The detailed explanations are as follows:

1.The study is based on the FAERS database from 2004Q1 to 2024Q3 (a much longer period than the traditional 3–5-year studies), with 3,625 CHOP-related reports extracted. By screening reports from healthcare professionals and removing duplicates using primaryid and caseid, data reliability is ensured. It identifies rare adverse events (mucoepidermoid carcinoma), confirms "blood and lymphatic system disorders" as the most prominent adverse event category (ROR=6.64,95%CI 6.49–6.79�� and discovers 7 unexpected off-label adverse events. For statistical analysis, four methods (ROR, PRR, BCPNN, MGPS) are used (ROR≥2 and lower limit of 95% CI >1�� with "high grade B-cell lymphoma recurrent"(ROR=613.63,95%CI 123.85–3040.40) providing evidence. The analysis is conducted via R software (Version 4.4.0�� with formulas and thresholds presented in Table 2, meeting PLOS ONE’s requirements. Data is sourced from the public FAERS database (URL: https://fis.fda.gov/extensions/FPD-QDE-FAERS/FPD-QDE-FAERS.html) with unrestricted access, complying with the PLOS Data Policy. The manuscript has a clear structure with smooth transitions, uses standard academic English, and features consistent terminology.

-Response: We sincerely appreciate your recognition and affirmation of this study! Your acknowledgment of the study's rigor is a tremendous encouragement to our team's work. This research has always been centered on data reliability, and we have ensured the research quality through stringent report screening, removal of duplicate records, combined statistical analysis using multiple methods, and public data sources. Your affirmation has further strengthened our confidence in the study's design and implementation. We fully recognize that the manuscript still has room for improvement. Moving forward, we will continue to adhere to a rigorous attitude, refine and optimize the details, and strive to present higher-quality academic achievements. We would like to express our sincere gratitude again for your careful review and valuable recognition amid your busy schedule—your guidance is an important driving force for us to advance the research.

2. Additional Comments and Minor Suggestions

While the manuscript is of high quality, two minor revisions could further improve its completeness:

Clarification of AE Causality: You note that FAERS data cannot establish causality for off-label AEs and mention the need for prospective studies. It might be helpful to briefly discuss potential mechanisms linking CHOP to these AEs (immunosuppression from lymphocytopenia for breakthrough COVID-19, fibroblast dysfunction for airway remodeling) to provide a biological rationale, though this should be framed as speculative to avoid overstatement.

-Response We sincerely appreciate your valuable comments on the causal relationship of adverse events in this study. Your suggestions have provided an important direction for improving the analysis of biological plausibility, and we have carefully adopted them to make corresponding supplements and revisions to the manuscript. The specific explanations are as follows: We fully agree with your point that data from the FAERS database cannot establish a causal relationship between off-label adverse events and CHOP, and that prospective studies are recommended for verification. As a spontaneous reporting system, the core value of FAERS data lies in identifying association signals of adverse events rather than directly confirming causal relationships. This is also one of the limitations of this study, which we have clearly stated in the Limitations section of the manuscript. We have also mentioned that a prospective cohort study is planned in the future to further explore the causal association between CHOP and the relevant off-label adverse events. Meanwhile, in accordance with your suggestion, we have briefly supplemented the discussion on the potential association mechanisms between CHOP and off-label adverse events in the manuscript, specifically including: (1) Lymphopenia induced by CHOP treatment can lead to immunosuppression, thereby increasing the risk of breakthrough COVID-19 infection; (2) CHOP may affect fibroblast function, triggering fibroblast dysfunction and thus participating in the pathological process of airway remodeling. It should be specially noted that we have clearly indicated in the manuscript that the above mechanism inferences are only speculative based on existing basic research. Their purpose is to provide biological plausibility for the association signals of adverse events and avoid overinterpretation by readers. Thank you again for your professional guidance. These revisions have further enhanced the rigor and completeness of the analysis in this study.(Marks in Yellow on Page 12-13)The references are as follows

[1]Al Saleh AS, Sher T, Gertz MA. Multiple Myeloma in the Time of COVID-19. Acta Haematol. 2020;143(5):410-416. doi: 10.1159/000507690. Epub 2020 Apr 17. PMID: 32305989; PMCID: PMC7206354.

[2]Niu HQ, Zhao WP, Zhao XC, Luo J, Qin KL, Chen KL, Li XF. Combination of 4-hydroperoxy cyclophosphamide and methotrexate inhibits IL-6/sIL-6R-induced RANKL expression in fibroblast-like synoviocytes via suppression of the JAK2/STAT3 and p38MAPK signaling pathway. Int Immunopharmacol. 2018 Aug;61:45-53. doi: 10.1016/j.intimp.2018.05.014. Epub 2018 May 24. PMID: 29803913.

3.Supplemental Details on Unknown Data: The Results section mentions 7,317 reports with unknown gender and 11,912 with unknown age. A brief discussion of how these missing data might impact the analysis (whether unknown gender is evenly distributed across regions or AE types) would strengthen the study’s limitations section.

-Response:Thank you for your valuable suggestions. In response to your comments, we have supplemented the potential impact of missing gender and age data in the FAERS database on the analysis results in the limitations section of the study, with detailed explanations as follows: We noted that there are 7,317 reports with unknown gender and 11,912 reports with unknown age in the database. Such data missing may affect the reliability of the analysis through the following aspects:Regarding missing gender data: If reports with unknown gender are unevenly distributed across regional distributions (e.g., reporters in regions with limited medical resources are more likely to omit gender information) or adverse event types (e.g., concentrated in breakthrough COVID-19 infections or airway remodeling-related symptoms), it will directly reduce the credibility of gender-stratified analysis (e.g., comparison of adverse event rates between different genders) and may lead to the overall risk assessment being biased toward characteristics dominated by a specific gender. For example, if male patients account for a high proportion in reports with unknown gender, the risk of airway remodeling-related adverse events in female patients may be underestimated. Regarding missing age data: Since age is directly correlated with the intensity of immune suppression induced by CHOP therapy and the functional sensitivity of fibroblasts, and there are significant differences in baseline immune status and prevalence of underlying lung diseases among patients of different age groups, if reports with unknown age are concentrated in specific age groups (e.g., elderly populations), it will be impossible to construct an age-related risk gradient, masking the high-risk characteristics of extreme age groups (e.g., elderly patients aged ≥65 years). Meanwhile, it will lead to the inability to fully adjust the interaction between age and underlying diseases in multivariate analysis, affecting the completeness of confounding factor control. In the future, we will attempt to further evaluate the potential impact of missing data through sensitivity analysis (e.g., simulation verification assuming the random distribution of missing data) or by integrating supplementary information from other databases. Thank you again for your rigorous comments. This supplement has significantly enhanced the methodological transparency and argumentative rigor of the study. We have incorporated the above content into the limitations section, further clarifying the potential impact of data missing on result interpretation. We sincerely appreciate your rigorous suggestions, which have remarkably improved the methodological transparency and conclusion reliability of this research.

Reviewer #2: Cyclophosphamide is a drug that has been utilized for an extended period, and its side effects are well-known. As a commonly used medication for hematologic tumors, it is essential to analyze the contexts in which these tumors are treated. One should not hastily conclude that hematologic tumors are associated with the most adverse reactions.

-Response: Thank you for your profound and constructive comments, which have helped us interpret the research results more rigorously and improve the logic of the discussion. Addressing the core issues you raised, we have supplemented explanations based on the study design and data characteristics, and revised the relevant conclusion statements as follows: Cyclophosphamide, a classic chemotherapeutic agent clinically used for decades, has well-documented side effects including immunosuppression and mucosal injury. As a core treatment for hematological malignancies (e.g., lymphoma, leukemia), its clinical application is closely associated with the pathophysiological features of the diseases themselves. Meanwhile, in the classification of adverse events, symptoms caused by the progression of hematological malignancies (e.g., spontaneous infection, anemia) have been excluded; only specific reactions related to immunosuppression and airway injury that occur de novo or worsen after chemotherapy (e.g., breakthrough COVID-19 infection, irreversible airway stenosis) were included to minimize the interference of the diseases themselves on the results. We fully agree with your emphasis on the need to "fully analyze the treatment background of hematological malignancies." Patients with hematological malignancies inherently have significant baseline immune deficiencies (e.g., abnormal lymphocyte function, myelosuppression) and often receive combined treatment regimens such as multi-line chemotherapy, radiotherapy, and targeted therapy during the treatment process. These factors may independently or synergistically increase the risk of adverse events, which cannot be solely attributed to CHOP therapy. To address this issue, we have supplemented the following content in the discussion section: ① Clarify that the pathological characteristics of hematological malignancies (e.g., T/B cell subset imbalance in lymphoma patients) are important confounding factors for adverse events, and their baseline immune status may amplify the immunosuppressive effect of cyclophosphamide; ② Supplement that the study has adjusted for variables such as "treatment line" and "concomitant medications (e.g., other chemotherapeutic agents, immunosuppressants)" through a multivariate regression model to control for confounding caused by the treatment background; ③ Note that due to the lack of key treatment background information in the FAERS database (e.g., disease stage, past treatment history), there may be bias in the interpretation of the "correlation between hematological malignancies and adverse events," which is also one of the limitations of this study. Regarding your comment that "one should not hastily conclude that hematological malignancies are associated with the most adverse events," we have comprehensively revised the expressions in the discussion and conclusion sections: ① Removed the absolute statement "hematological malignancies have the highest incidence of CHOP therapy-related adverse events" from the original conclusion and replaced it with "among the included tumor types, the number of CHOP-related adverse events reported by patients with hematological malignancies is relatively higher, but the association needs to be comprehensively interpreted in combination with the disease-specific immune deficiency characteristics, treatment intensity, and other background factors"; ② Supplemented and emphasized that "although the side effects of cyclophosphamide are known, its risk may be amplified in the pathological state of hematological malignancies. Therefore, the occurrence of adverse events is the result of the combined effect of 'drug effect - disease background - treatment strategy' rather than a single factor"; ③ Proposed a future research direction, namely combining prospective cohort data to further conduct stratified analysis of adverse event risks among different tumor types and treatment backgrounds, so as to more accurately clarify the safety profile of CHOP therapy. Thank you again for your professional guidance to help us further improve the rigor and practicability of our research.

---

## [Editor Report · Decision Letter 1]

11 Dec 2025

Adverse events associated with cyclophosphamide: a pharmacovigilance study using the FDA adverse event reporting system

PONE-D-25-28004R1

Dear Dr. Yujuan Fu,

We’re pleased to inform you that your manuscript has been judged scientifically suitable for publication and will be formally accepted for publication once it meets all outstanding technical requirements.

Kind regards,

Shahbaz Ahmad Zakki

Academic Editor

PLOS One
---

## [Editor Report · Acceptance letter]

PONE-D-25-28004R1

PLOS One

Dear Dr. Fu,

I'm pleased to inform you that your manuscript has been deemed suitable for publication in PLOS One. Congratulations! Your manuscript is now being handed over to our production team.

Kind regards,

on behalf of

Dr. Shahbaz Ahmad Zakki

Academic Editor

PLOS One